materials science/mechanical engineering/ structural engineering

periodic framework, tetrahedron, vibration, auxeticity, zero stiffness, mechanical metamaterial

**Author for correspondence:**
H. Tanaka
e-mail: htanaka@mech.eng.osaka-u.ac.jp

# Auxetic vibration behaviours of periodic tetrahedral units with a shared edge

## H. Tanaka[1], S. Asao[1] and Y. Shibutani[1,2]

[1]Department of Mechanical Engineering, Osaka University, 2-1 Yamadaoka, Suita, Osaka 565-0871, Japan
[2]Nanotechnology Program, VNU Vietnam Japan University, Luu Huu Phuoc Street, My Dinh 1 Ward, Nam Tu Liem District, Ha Noi, Viet Nam

HT, 0000-0002-1713-2734

A very low-frequency mode supported within an auxetic structure is presented. We propose a constrained periodic framework with corner-to-corner and edge-to-edge sharing of tetrahedra and develop a kinematic model incorporating two types of linear springs to calculate the momentum term under infinitesimal transformations. The modal analysis shows that the microstructure with its two degrees of freedom has both low- and high-frequency modes under auxetic transformations. The low-frequency mode approaches zero frequency when the corresponding spring constant tends to zero. With regard to coupled eigenmodes, the stress–strain relationship of the uniaxial forced vibration covers a wide range. When excited, a very slow motion is clearly observed along with a structural expansion for almost zero values of the linear elastic modulus.

## 1. Introduction

Various repetitive structures composed of simple geometric shapes have been investigated extensively with the endeavour to enhance fundamental properties such as rigidity and flexibility [1–4]. While many rigid microstructures based on trusses have been developed from a mechanical viewpoint [5,6], flexible microstructures are expected to realize anomalous mechanical characteristics in solid matter, being distinctive in having, for example, non-positive values of Poisson's ratio [7–11] or coefficient of thermal expansion [12–14].

The two-dimensional corner-linked frameworks of polygons such as triangles and squares have been developed by employing several types of flexible mechanisms per unit cell [15–18]. In three-dimensional frameworks, a tetrahedron unit is one possible geometrical component; indeed, a subgroup of synthesized tetrahedra may be designed and used at a microscopic scale in chemistry [19]. For the most part, corner-shared structures are treated despite other types of polyhedral joining, i.e. edge-to-edge

(two-dimensional) and face-to-face (three-dimensional) connections. The potential mechanisms enriched through these three-dimensional connections are less discussed.

Auxetic models across different geometrical groups may be categorized into classes based on their deformation mechanisms [20]. One fundamental auxetic mechanism is the rotation of rigid units such as squares and rectangles in two dimensions [21,22] and tetrahedra in three dimensions [23,24]. For the latter, experimental and numerical reports on polycrystalline solids have indicated that the α- and β-phases of cristobalite structures are composed of $SiO_2$, in which all the corners of the tetrahedra $(SiO_4)$ are shared, and they exhibit auxetic behaviours in a state at a particular temperature [25,26]. Recently, computations have predicted several types of polycrystal materials having negative values for the directional and/or the homogeneous Poisson ratios that arise through tetrahedral rotations [27].

Corner-linked tetrahedra potentially have a specific vibration property. When rigid regions percolate but the kinetic degrees of freedom are finite inside, low frequency vibrations are excited; for example, in some silicate crystals such as β-cristobalite low-frequency modes have been demonstrated in molecular simulations implementing a bond/stretching force potential [28,29]. The force-free coordinated rotation of tetrahedra about their corners play an important role in soft vibrations. Although these low-frequency modes are not simple because the tetrahedra are distorted, the fundamental mechanism can be modelled by a rigid rotation of tetrahedra.

In this context, we present a tetrahedral structure with strict restrictions that permits two transformation mechanisms; specifically, an auxetic mode and a low-frequency mode. We focus on edge sharing of neighbouring tetrahedra and develop a framework with their point- and line-wise pivotal connections. This mixed connectivity allows the spring-interacted tetrahedra to adopt two distinct mechanical properties. In our previous study, we proposed a class of constrained periodic polyhedral structures with two degrees of freedom and developed a static model involving interacting spring elements. Linear and nonlinear transformation analyses clarify why Poisson's ratio of these structures is negative and undergoes a three-dimensional coordinated rotation of edge-shared tetrahedra [30]. Additionally, the structure exhibits zero stiffness at the initial configuration when the spring coefficients satisfy a specific condition. In this study, we extend the static model of the kinetic structure by associating a rotational momentum term for each tetrahedron unit.

In §2, we describe our tetrahedral framework and the manner in which each unit cell with its two degrees of freedom undergoes its uniform transformation, including the infinitesimal displacements yielding auxeticity. In §3, we formulate the stiffness and kinetic motion of the interactional structure assuming an infinitesimal tetrahedral rotation. In §4, we perform a modal analysis, which indicates that the periodically constrained structure has mixed vibrational characteristics, specifically the zero-frequency mode of the auxetic transformation. We then perform a frequency response analysis for the uniaxial forced vibration and verify similar modal features from the stress–strain relationship. Last, in §5, we summarize the bi-functional structure.

# 2. Polyhedral units and interactional transformation

## 2.1. Structural modelling

We consider a periodic polyhedral structure, the unit cell of which is made up of eight regular tetrahedra that are paired up. These four pairs share four edges forming a square linkage. The remaining vertices are connected pivotally with those in adjacent unit cells in an orthogonal periodic fashion. The periodic frameworks have over-constrained mechanisms with the states of self-stress and, assuming uniformity, a unit cell undergoes a continuous affine transformation having two degrees of freedom, here called the *bimode* transformation [30].

As shown in figure 1a, the initial configuration of the unit cell forms a *stella octangula* [31,32]; its cell length is denoted $2\ell$. The bimode transformation is described by rotation parameter $\theta$ and elongation rate $\gamma$, which will be defined below. Figure 1b illustrates the representative transformation with the change in $\theta$ by holding $\gamma = 1$, specially called the *unimode* transformation.

The analytical model of the periodic polyhedral structure with spring interactions (figure 2a) has eight structural elements, the first of which is composed of a single tetrahedral unit ABCD connected to the other tetrahedra and interacts with them via three different linear springs, with spring constants $k_B$, $k_C$ and $k_D$: another connected joint reflected in the corresponding plane. In the analytical model, the joints B and C are constrained to move along the $x_1$-axis and $x_3$-axis, respectively. For the inserted spring at B along OB, its length being $2\gamma\ell$, where $\gamma$ is the elongation rate (figure 2b). The two springs

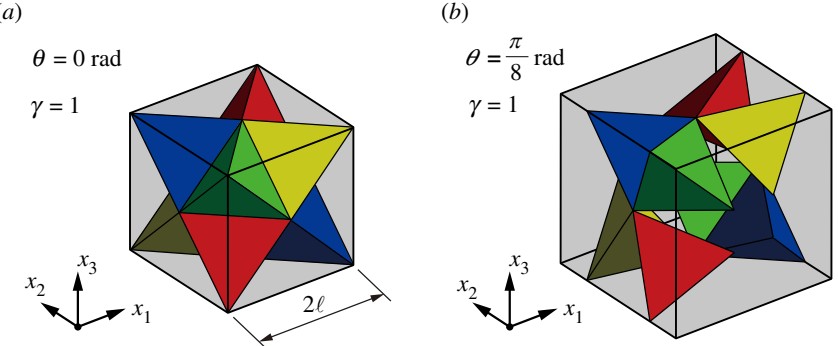

**Figure 1.** Structural model involving eight connected tetrahedra per unit cell: three-dimensional views of (a) initial and (b) transformed configurations fixed at $\gamma = 1$.

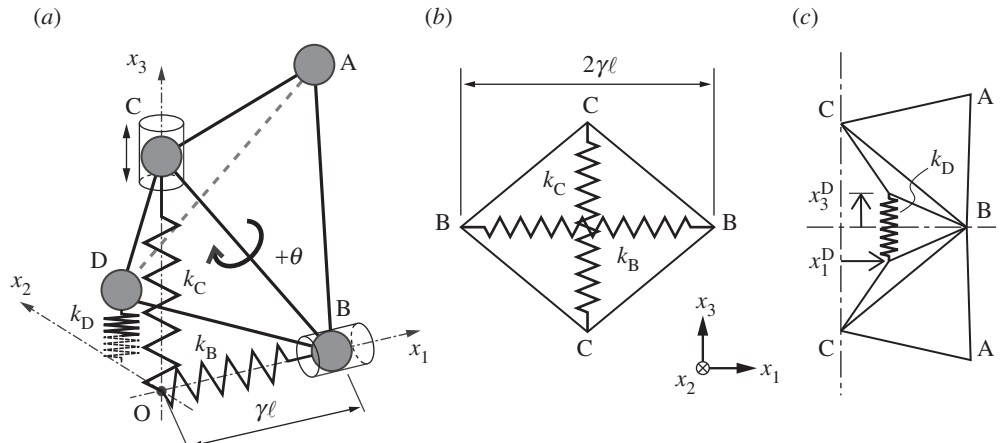

**Figure 2.** (a) Schematic of the first of eight structural elements with interacting springs of spring constants $k_B$, $k_C$ and $k_D$. The tetrahedral configuration is determined by two geometric parameters: the elongation rate $\gamma$ and the angle $\theta$ through which ABCD rotates about BC; (b,c) the allocated positions of the linear springs with $k_B$, $k_C$ and $k_D$. Note that $x_1^D$ and $x_3^D$ are the positions of the joint D in the $x_1$- and $x_3$-directions.

pass through the origin because of the transformation symmetry. The vertical spring along the $x_3$-axis is connected with an adjacent tetrahedron, which is a reflection of the first tetrahedron about the $(x_1, x_2)$-plane (figure 2c).

## 2.2. Formulation of the transformation

The bimode transformation decomposes into two distinct tetrahedral rotations: one is a rotation by $\theta$ of ABCD about the shared edge (link BC in figure 2), and the other is a rotation constrained on the $(x_1, x_3)$-plane. Figure 3 illustrates the latter rotation, induced by the extension/contraction of the two linear springs along OB and OC. The parameters $\gamma$ and $\tau$ denote the rates of elongation of springs OB and OC, with $\tau = \sqrt{2 - \gamma^2}$ from which $\partial\gamma/\partial\tau = -\tau/\gamma$ obtains. The square A'B'C'D' is the projection of the tetrahedron onto the $(x_1, x_3)$-plane along with changes in $\gamma$ and $\tau$. The expressions for the positions of the four vertices are

$$\boldsymbol{x}^{A'} = \ell \begin{pmatrix} \dfrac{\gamma^2 - 1}{\gamma - \tau} \\ 1 \\ \dfrac{\gamma^2 - 1}{\gamma - \tau} \end{pmatrix}, \quad \boldsymbol{x}^{B'} = \begin{pmatrix} \gamma\ell \\ 0 \\ 0 \end{pmatrix}, \quad \boldsymbol{x}^{C'} = \begin{pmatrix} 0 \\ 0 \\ \tau\ell \end{pmatrix}, \quad \boldsymbol{x}^{D'} = \ell \begin{pmatrix} \dfrac{1 - \gamma\tau}{\gamma - \tau} \\ 1 \\ -\dfrac{1 - \gamma\tau}{\gamma - \tau} \end{pmatrix}. \tag{2.1}$$

The unit vector $\boldsymbol{n}$ from point C' to point B' may be expressed in the form

$$\boldsymbol{n} \equiv (n_1, n_2, n_3)^{\mathsf{T}} = \frac{\boldsymbol{C'B'}}{|\boldsymbol{C'B'}|} = \frac{1}{\sqrt{2}}(\gamma, 0, -\tau)^{\mathsf{T}}. \tag{2.2}$$

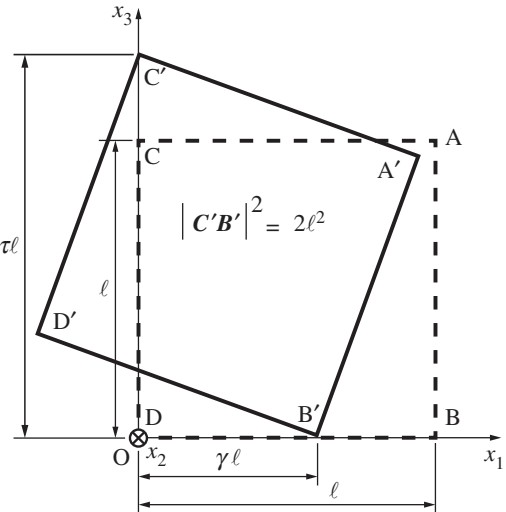

**Figure 3.** Schematic of the regular tetrahedral rotation constrained to the $(x_1, x_3)$-plane.

According to Rodrigues' formula, the matrix that represents a rotation about $\boldsymbol{n}$ by an angle $\theta$ may be expressed as

$$\boldsymbol{R} = \boldsymbol{I} + \sin\theta \boldsymbol{N} + (1 - \cos\theta)\boldsymbol{N}^2 \tag{2.3}$$

$$= \begin{pmatrix} \dfrac{\gamma^2 + \tau^2\cos\theta}{2} & \dfrac{\tau\sin\theta}{\sqrt{2}} & -\dfrac{\gamma\tau(1-\cos\theta)}{2} \\[2mm] -\dfrac{\tau\sin\theta}{\sqrt{2}} & \cos\theta & -\dfrac{\gamma\sin\theta}{\sqrt{2}} \\[2mm] -\dfrac{\gamma\tau(1-\cos\theta)}{2} & \dfrac{\gamma\sin\theta}{\sqrt{2}} & \dfrac{\tau^2 + \gamma^2\cos\theta}{2} \end{pmatrix}, \tag{2.4}$$

where $\boldsymbol{N}$ denotes the cross-product matrix,

$$\boldsymbol{N} = \begin{pmatrix} 0 & -n_3 & n_2 \\ n_3 & 0 & -n_1 \\ -n_2 & n_1 & 0 \end{pmatrix}. \tag{2.5}$$

Using equations (2.1) and (2.4), the transformed tetrahedron ABCD in the bimodal structure may be represented vectorially as

$$\boldsymbol{x}^i = \boldsymbol{R}(\theta, \gamma, \tau)(\boldsymbol{x}^{i'} - \boldsymbol{x}^{C'}) + \boldsymbol{x}^{C'}, \quad i = \text{A, B, C, D}. \tag{2.6}$$

The position vectors of nodes A and D, denoted $\boldsymbol{x}^A$ and $\boldsymbol{x}^D$, determine the unit cell after the transformation. From equation (2.6), their expressions are

$$\boldsymbol{x}^A = \ell \begin{pmatrix} \dfrac{\tau\cos\theta}{2} + \dfrac{\tau\sin\theta}{\sqrt{2}} + \dfrac{\gamma}{2} \\[2mm] \cos\theta - \dfrac{\sin\theta}{\sqrt{2}} \\[2mm] \dfrac{\gamma\cos\theta}{2} + \dfrac{\gamma\sin\theta}{\sqrt{2}} + \dfrac{\tau}{2} \end{pmatrix} \tag{2.7}$$

and

$$\boldsymbol{x}^D = \ell \begin{pmatrix} -\dfrac{\tau\cos\theta}{2} + \dfrac{\tau\sin\theta}{\sqrt{2}} + \dfrac{\gamma}{2} \\[2mm] \cos\theta + \dfrac{\sin\theta}{\sqrt{2}} \\[2mm] -\dfrac{\gamma\cos\theta}{2} + \dfrac{\gamma\sin\theta}{\sqrt{2}} + \dfrac{\tau}{2} \end{pmatrix}. \tag{2.8}$$

Let $X_{i\in\{1,2,3\}}$ denote the three half-lengths of the sides of the transformed unit cell given by

$$X_1 = x_1^A, \quad X_2 = x_2^D \quad \text{and} \quad X_3 = x_3^A. \tag{2.9}$$

Therefore, the effective strains that are applied to the unit cell are

$$\varepsilon_i^* = \frac{X_i - \ell}{\ell}, \quad i = 1, 2, 3. \tag{2.10}$$

## 2.3. Infinitesimal transformation

Let $\epsilon$ be an infinitesimal parameter; the increments in the two variables associated with the bimode transformation are then

$$\theta = \theta_0 + \epsilon\Delta\theta \quad \text{and} \quad \gamma = \gamma_0 + \epsilon\Delta\gamma, \tag{2.11}$$

where $\theta_0 = 0$ and $\gamma_0 = 1$ are the initial values. Using the second relation of equation (2.11), $\tau$ (>0) is expressed by

$$\tau = (1 - 2\epsilon\Delta\gamma + O(\epsilon^2))^{1/2} = 1 - \epsilon\Delta\gamma + O(\epsilon^2), \tag{2.12}$$

where O is Landau's symbol, meaning 'of order', the terms of which are ignored. The structural transformation then involves infinitesimal tetrahedral motions about $\Delta\theta$ and $\Delta\gamma$. By substituting cos $(\epsilon\Delta\theta) = 1 + O(\epsilon^2)$, sin $(\epsilon\Delta\theta) = \epsilon\Delta\theta + O(\epsilon^3)$, and equations (2.11) and (2.12) into equation (2.10), we have

$$\varepsilon_i^* = \frac{\sqrt{2}}{2}\epsilon\Delta\theta + O(\epsilon^2), \quad i = 1, 2, 3. \tag{2.13}$$

Therefore, the effective deformation per unit cell is determined by $\Delta\theta$, and is independent of $\Delta\gamma$. The structure exhibits expansion behaviour if $\Delta\theta > 0$ whereas it shrinks if $\Delta\theta < 0$.

# 3. Stiffness and tetrahedral motion

## 3.1. Energy description into stiffness per unit cell

The elastic energy of the structural model per unit cell is given by

$$U_e = \frac{1}{2}k_B(2(\gamma - 1)\ell)^2 + \frac{1}{2}k_C(2(\tau - 1)\ell)^2 + 2 \times \frac{1}{2}k_D(2x_3^D)^2. \tag{3.1}$$

Setting $k_B = k_C \equiv k$, we derive the linear approximation of equation (3.1) when the system is subjected to an infinitesimal transformation. From equation (2.8), the vertical displacement undergone by the spring connected to D is written as

$$2x_3^D = \epsilon\ell\left(\sqrt{2}\Delta\theta - 2\Delta\gamma\right) + O(\epsilon^2). \tag{3.2}$$

Taking into account only the first-order terms in $\epsilon$ and substituting equations (2.12) and (3.2) into equation (3.1), the elastic energy becomes

$$U_e \simeq \epsilon^2\left[4k\ell^2(\Delta\gamma)^2 + k_D\ell^2\left(\sqrt{2}\Delta\theta - 2\Delta\gamma\right)^2\right].$$

Setting $\epsilon\Delta\theta \to \theta$ and $\epsilon\Delta\gamma \to \varphi$ in the above equation, we obtain

$$U_e \simeq 4k\ell^2\varphi^2 + k_D\ell^2\left(\sqrt{2}\theta - 2\varphi\right)^2. \tag{3.3}$$

Hence, the unit cell stiffness with respect to $\theta$ and $\varphi$ is the Hessian matrix of

$$\boldsymbol{K} = \begin{pmatrix} \dfrac{\partial^2 U_e}{\partial\theta^2} & \dfrac{\partial^2 U_e}{\partial\varphi\partial\theta} \\ \dfrac{\partial^2 U_e}{\partial\theta\partial\varphi} & \dfrac{\partial^2 U_e}{\partial\varphi^2} \end{pmatrix} = 4k_D\ell^2\begin{pmatrix} 1 & -\sqrt{2} \\ -\sqrt{2} & 2\kappa + 2 \end{pmatrix}, \tag{3.4}$$

where $\kappa \equiv k/k_D$ denotes the non-dimensional spring constant.

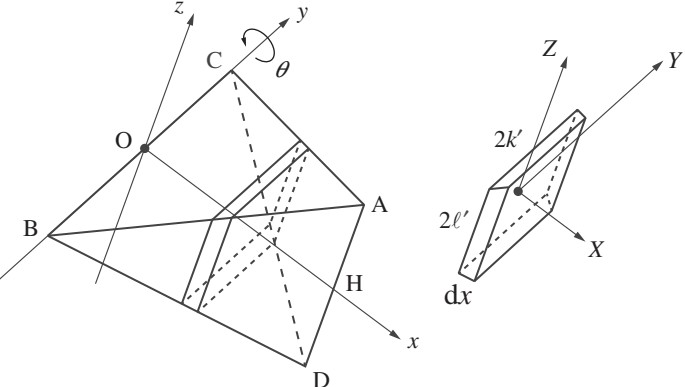

**Figure 4.** Tetrahedron unit rotating about the directional vector from point C to point B (the y-axis).

## 3.2. Rotational inertia of the tetrahedron unit

Let $I_\theta$ and $I_\varphi$ be the moments of inertia of a single tetrahedron with respect to $\theta$ and $\varphi$, respectively. Their expressions are

$$I_\theta = \frac{7}{60}\rho\ell^5 \quad \text{and} \quad I_\varphi = \frac{1}{5}\rho\ell^5, \tag{3.5}$$

where $\rho$ is the density. The derivations are detailed in §§3.2.1 and 3.2.2.

### 3.2.1. Derivation of $I_\theta$

Establishing a Cartesian coordinate system O-$xyz$ for the regular tetrahedron ABCD (figure 4), we consider a rotation about line CB (the $y$-axis) by $\theta$. Let $a$ be the side length of the polyhedron. Geometrically, we have

$$OA = \frac{\sqrt{3}}{2}a \quad \text{and} \quad OH = a\sqrt{\frac{3}{4} - \frac{1}{4}} = \frac{1}{\sqrt{2}}a. \tag{3.6}$$

The rectangular area $S$ positioned at $x$ from the origin is given by $S = 4k'\ell'$ (see inset of figure 4). The mass of the rectangle per unit thickness is expressed as

$$m(x) = 4\rho k'(x)\ell'(x). \tag{3.7}$$

Geometric similarity implies the length ratio condition, $2k' : (OH - x) = BC : OH = a : a/\sqrt{2}$. Therefore,

$$k' = \frac{1}{\sqrt{2}}\left(\frac{a}{\sqrt{2}} - x\right). \tag{3.8}$$

In a similar manner,

$$\ell' = \frac{1}{\sqrt{2}}x, \tag{3.9}$$

obtained from the geometric similarity of $(OH - x) : 2\ell' = OH : a$.

We next set new coordinates O'-$XYZ$ at the centre of gravity of the plate (figure 5). The moments of inertia about the $Y$- and $Z$-axes are then

$$I_Y = \int_{-\ell'}^{\ell'} \rho(2k')Z^2 dZ = 2\rho k'\left[\frac{1}{3}Z^3\right]_{-\ell'}^{\ell'} = \frac{4}{3}\rho k'\ell'^3 \left(= \frac{1}{3}m\ell'^2\right) \tag{3.10}$$

and

$$I_Z = \int_{-k'}^{k'} \rho(2\ell')Y^2 dY = 2\rho\ell'\left[\frac{1}{3}Y^3\right]_{-k'}^{k'} = \frac{4}{3}\rho k'^3\ell' \left(= \frac{1}{3}mk'^2\right). \tag{3.11}$$

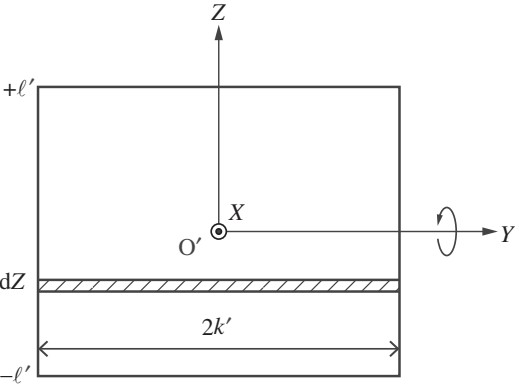

**Figure 5.** Moment of inertia of the plate about the Y-axis.

In the terms of the rotation about the X-axis,

$$I_X = \int_S (Y^2 + Z^2)\,\mathrm{d}Y\,\mathrm{d}Z = I_Y + I_Z = \frac{1}{3}m(k'^2 + \ell'^2). \tag{3.12}$$

From equations (3.7)–(3.9) and (3.11), the rotational inertia associated with coordinate $\theta$, denoted $I_\theta$, follows:

$$
\begin{aligned}
I_\theta &= \int_0^{a/\sqrt{2}} (I_Y(x) + m(x)x^2)\,\mathrm{d}x \\
&= \int_0^{a/\sqrt{2}} \left(\frac{1}{3}m\ell'^2 + mx^2\right)\mathrm{d}x \\
&= \rho \int_0^{a/\sqrt{2}} \left\{\frac{x}{12}\left(\frac{a}{\sqrt{2}} - x\right)^3 + 2x^3\left(\frac{a}{\sqrt{2}} - x\right)\right\}\mathrm{d}x \\
&= \frac{7}{240\sqrt{2}}\rho a^5.
\end{aligned}
$$

With $a = \sqrt{2}\ell$, we obtain the first relation in equation (3.5).

### 3.2.2. Derivation of $I_\varphi$

Following the procedures used in equations (2.11) and (2.12), we approximate $\tau = \sqrt{2 - \gamma^2}$ by applying the binomial expansion formula and considering terms up to second order in $\epsilon$. Therefore,

$$
\begin{aligned}
\tau &= (1 - 2\epsilon\Delta\gamma - (\epsilon\Delta\gamma)^2)^{1/2} \\
&= 1 + \frac{1}{2}(-2\epsilon\Delta\gamma - (\epsilon\Delta\gamma)^2) - \frac{1}{8}(-2\epsilon\Delta\gamma)^2 + O(\epsilon^3) \\
&\simeq 1 - \epsilon\Delta\gamma - \epsilon^2(\Delta\gamma)^2. \tag{3.13}
\end{aligned}
$$

Substituting $\gamma = 1 + \epsilon\Delta\gamma$ and equation (3.13) into equation (2.1), we have

$$
x^{A'} = \ell\begin{pmatrix} 1 \\ 1 \\ 1 \end{pmatrix}, \quad
x^{B'} = \ell\begin{pmatrix} 1 + \epsilon\Delta\gamma \\ 0 \\ 0 \end{pmatrix}, \quad
x^{C'} = \ell\begin{pmatrix} 0 \\ 0 \\ 1 - \epsilon\Delta\gamma - \epsilon^2(\Delta\gamma)^2 \end{pmatrix}
$$

and

$$
x^{D'} = \ell\begin{pmatrix} \dfrac{1 - (1 + \epsilon\Delta\gamma)(1 - \epsilon\Delta\gamma - \epsilon^2(\Delta\gamma)^2)}{(1 + \epsilon\Delta\gamma) - (1 - \epsilon\Delta\gamma - \epsilon^2(\Delta\gamma)^2)} \\ 1 \\ -\dfrac{1 - (1 + \epsilon\Delta\gamma)(1 - \epsilon\Delta\gamma - \epsilon^2(\Delta\gamma)^2)}{(1 + \epsilon\Delta\gamma) - (1 - \epsilon\Delta\gamma - \epsilon^2(\Delta\gamma)^2)} \end{pmatrix} = \ell\begin{pmatrix} \epsilon\Delta\gamma \\ 1 \\ -\epsilon\Delta\gamma \end{pmatrix}.
$$

Ignoring terms of order $\epsilon^2$ and replacing $\epsilon\Delta\gamma$ with $\varphi$, the vertex positions after the tetrahedron is

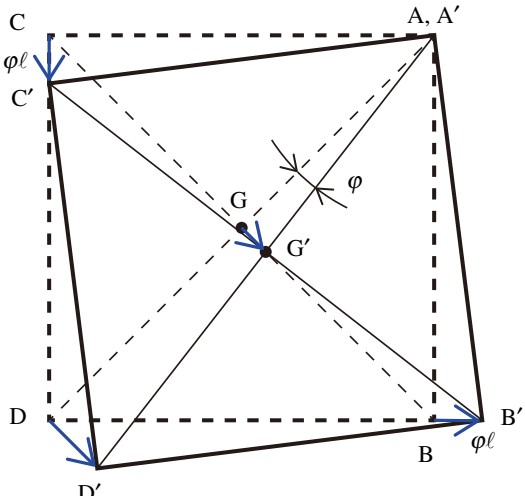

**Figure 6.** Infinitesimal rotation by $\varphi$ about point A.

slid infinitesimally along the $x_1$-axis are given as

$$x^{\mathrm{A}'} = \ell \begin{pmatrix} 1 \\ 1 \\ 1 \end{pmatrix}, \quad x^{\mathrm{B}'} = \ell \begin{pmatrix} 1 + \varphi \\ 0 \\ 0 \end{pmatrix} \tag{3.14}$$

and

$$x^{\mathrm{C}} = \ell \begin{pmatrix} 0 \\ 0 \\ 1 - \varphi \end{pmatrix}, \quad x^{\mathrm{D}'} = \ell \begin{pmatrix} \varphi \\ 1 \\ -\varphi \end{pmatrix}. \tag{3.15}$$

Figure 6 illustrates the trajectory of the tetrahedron ABCD to A′B′C′D′, represented by equations (3.14) and (3.15). The schematic shows that the sliding motion, projected onto the $(x_1, x_3)$-plane, is equivalent to an infinitesimal polyhedral rotation about the axis passing through point $A$ parallel to the $x_2$-axis. Because the rotation axis is along the $X$-axis (figure 4), from equations (3.7)–(3.9), (3.12), and $\mathrm{AH} = a/2$, we may derive the rotational inertia concerning $\varphi$,

$$\begin{aligned} I_\varepsilon &= \int_0^{a/\sqrt{2}} (I_X + m(x)(\mathrm{AH})^2)\,\mathrm{d}x \\ &= \int_0^{a/\sqrt{2}} \frac{4}{3}\rho k' \ell'(k'^2 + \ell'^2)\mathrm{d}x + \int_0^{a/\sqrt{2}} a^2 \rho k' \ell'\,\mathrm{d}x \\ &= \frac{1}{120\sqrt{2}}\rho a^5 + \frac{1}{24\sqrt{2}}\rho a^5 \\ &= \frac{1}{20\sqrt{2}}\rho a^5. \end{aligned}$$

With $a = \sqrt{2}\ell$, we obtain the second relation of equation (3.5).

# 4. Vibration characteristics on the structural model

## 4.1. Modal analyses

For the unloaded structure, the equation of motion for the bimode transformation of a unit cell is given by

$$M\ddot{u} + Ku = 0, \tag{4.1}$$

where $u = (\theta, \varphi)^\mathsf{T}$, $0 = (0, 0)^\mathsf{T}$, $\ddot{u} \equiv \mathrm{d}^2 u/\mathrm{d}t^2$, $M$ is the generalized mass matrix described as

$$M = \begin{pmatrix} 8I_\theta & 0 \\ 0 & 8I_\varphi \end{pmatrix}. \tag{4.2}$$

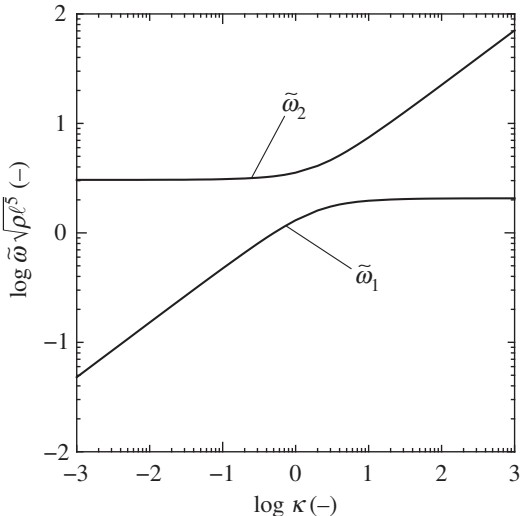

**Figure 7.** Relationship between $\tilde{\omega}$ and $\kappa$ for the low and high angular frequency modes. The solid lines mark analytical solutions of equation (4.6), denoted $\tilde{\omega}_1$ and $\tilde{\omega}_2$ ($\tilde{\omega}_1 < \tilde{\omega}_2$). Note that $\tilde{\omega}\sqrt{\rho \ell^5} = \omega\sqrt{\rho \ell^3 / k_{\mathrm{D}}}$ is dimensionless.

The self-excited vibration with positive angular frequency $\omega > 0$ is expressed as $\boldsymbol{u} = \boldsymbol{a}\mathrm{e}^{\mathrm{i}\omega t}$, where i is the imaginary unit and $\boldsymbol{a} \in \mathbb{R}^2$. Substituting $\boldsymbol{u}$ into equation (4.1), we obtain the characteristic equation,

$$\boldsymbol{K} - \omega^2 \boldsymbol{M} = \boldsymbol{0}. \tag{4.3}$$

Putting $\omega^2 = k_{\mathrm{D}}\ell^2 \tilde{\omega}^2$ and substituting equations (3.4) and (4.2), the left-hand side matrix in equation (4.3) becomes

$$4k_{\mathrm{D}}\ell^2 \begin{pmatrix} 1 - 2I_\theta\tilde{\omega}^2 & -\sqrt{2} \\ -\sqrt{2} & 2\kappa + 2 - 2I_\varphi\tilde{\omega}^2 \end{pmatrix} \equiv \boldsymbol{F}. \tag{4.4}$$

The system of equations has a non-trivial solution if and only if $\det \boldsymbol{F} = 0$, that is,

$$2I_\theta I_\varphi \tilde{\omega}^4 - (2(\kappa + 1)I_\theta + I_\varphi)\tilde{\omega}^2 + \kappa = 0. \tag{4.5}$$

Solving equation (4.5), we find

$$\tilde{\omega}^2 = \frac{A \pm \sqrt{A^2 - 8\kappa I_\theta I_\varphi}}{4I_\theta I_\varphi}, \quad A = 2(\kappa + 1)I_\theta + I_\varphi. \tag{4.6}$$

Plotting $\tilde{\omega}$ versus $\kappa$ in equation (4.6), the two solid curves as shown in figure 7 correspond to the low and high angular frequencies denoted $\tilde{\omega}_1$ and $\tilde{\omega}_2$ ($\tilde{\omega}_1 < \tilde{\omega}_2$), where log indicates the common logarithm with base 10. The two angular frequencies have unique trends characterized by the limits taken for the non-dimensional spring constant, specifically, $\kappa \to 0$ and $\kappa \to \infty$.

When $\kappa = 0$, two distinct eigenvalues arise,

$$\tilde{\omega}_1 = 0 \quad \text{and} \quad \tilde{\omega}_2 = \pm\left(\frac{1}{2I_\theta} + \frac{1}{I_\varphi}\right)^{1/2}. \tag{4.7}$$

The corresponding frequency modes $\phi_{i\in\{1,2\}}$ to $\tilde{\omega}_{i\in\{1,2\}}$ become

$$\phi_1 = \begin{pmatrix} \sqrt{2} \\ 1 \end{pmatrix} \quad \text{and} \quad \phi_2 = \begin{pmatrix} 1 \\ -\sqrt{2}(I_\theta/I_\varphi) \end{pmatrix}. \tag{4.8}$$

By contrast, when $\kappa$ is sufficiently large, the dominant terms in the solution of equation (4.6) are

$$\tilde{\omega}^2 = \frac{A}{4I_\theta I_\varphi}\left(1 \pm \sqrt{1 - \frac{8\kappa I_\theta I_\varphi}{A^2}}\right) \simeq \left(\frac{\kappa + 1}{2I_\varphi} + \frac{1}{4I_\theta}\right)\left[1 \pm \left(1 - \frac{I_\varphi}{\kappa I_\theta}\right)\right]. \tag{4.9}$$

Therefore, the two eigenvalues and their corresponding eigenmodes become

$$\tilde{\omega}_1 = \pm\left(\frac{1}{2I_\theta}\right)^{1/2}, \quad \tilde{\omega}_2 = \pm\left(\frac{\kappa+1}{I_\varphi}\right)^{1/2} \tag{4.10}$$

and

$$\phi_1 = \begin{pmatrix} 1 \\ 0 \end{pmatrix}, \quad \phi_2 = \begin{pmatrix} 0 \\ 0 \end{pmatrix}. \tag{4.11}$$

For the low-frequency mode $\tilde{\omega}_1$, as $\kappa \to 0$ ($k_D \gg k$), the vibration behaviour $\phi_1$ corresponds to a zero-frequency mode because $\tilde{\omega}_1 = 0$ from equation (4.7). Under the infinitesimal transformation, we obtain the $x_1$-and $x_3$-axial positions of node D, expressed by

$$x_1^D \simeq \ell\left(\frac{\theta}{\sqrt{2}} + \varphi\right) \quad \text{and} \quad x_3^D \simeq \ell\left(\frac{\theta}{\sqrt{2}} - \varphi\right). \tag{4.12}$$

Hence, $x_1^D = 2\ell$ and $x_3^D = 0$ with $\phi_1 = (\sqrt{2}, 1)^\top$ in equation (4.8). From energy considerations of equation (3.1) with $x_3^D = 0$, a very slow auxetic vibration occurs that stores no elastic energy—the vibrated structure undertakes an auxetic/shrink motion, governed by $\theta$ as in equation (2.13). When $\kappa \to \infty$ ($k_D \ll k$), $x_1^D = \ell/\sqrt{2}$ and $x_3^D = \ell/\sqrt{2}$, from $\phi_1 = (1, 0)^\top$ in equation (4.11); hence, only a tetrahedral rotation by $\theta$ occurs as the unimode transformation. Note that, in both limits with $\theta < 0$, the structure never shrinks physically; indeed, the tetrahedral units make contact with each other because $x_1^D < 0$.

According to equations (3.5), (4.7), (4.8) and (4.12), when $\kappa = 0$, the mode with high angular frequency $\tilde{\omega}_2$ begins with a fixed value for which $x_1^D = -\sqrt{2}\ell/12$ and $x_3^D = 13\sqrt{2}\ell/12$. Hence, the structure is locked in a physical sense because $x_1^D < 0$ with $\theta > 0$; alternatively, $x_3^D < 0$ with $\theta < 0$. When $\kappa \to \infty$, $\tilde{\omega}_2$ diverges by equation (4.10) and the structure is fixed with $\phi_2 = (0, 0)^\top$.

## 4.2. Frequency response analysis

When the structure is expanded infinitesimally and with a force applied, the equation of motion of the unit cell from equation (4.1) is expressed as

$$\left.\begin{array}{l} M_{11}\ddot{\theta} + K_{11}\theta + K_{12}\varphi = q \\ M_{22}\ddot{\varphi} + K_{21}\theta + K_{22}\varphi = 0, \end{array}\right\} \tag{4.13}$$

and

where $q$ denotes the internal moment, and $M_{ij}$ and $K_{ij}$ denote the $ij$th component of $\boldsymbol{M}$ and $\boldsymbol{K}$, respectively. Note that the second equation of equation (4.13) never contributes to the structural expansion because the rotation by $\varphi$ about point A is projected onto the ($x_1$, $x_3$)-plane (see Fig. (6)). Recalling equation (2.13), the uniaxial strain $\varepsilon_1^*$ of interest is related to $\theta$ by $\varepsilon_1^* = \sqrt{2}\theta/2$. We then develop the forced vibration system per unit cell in the $x_1$-direction to generalize equation (4.13),

$$\begin{cases} M_{11}\ddot{\varepsilon}_1^* + K_{11}\varepsilon_1^* + \frac{1}{\sqrt{2}}K_{12}\varphi = \frac{q}{\sqrt{2}} \\ \frac{1}{\sqrt{2}}M_{22}\ddot{\varphi} + K_{21}\varepsilon_1^* + \frac{1}{\sqrt{2}}K_{22}\varphi = 0 \end{cases}$$
$$\Leftrightarrow \begin{cases} \frac{1}{8\ell^3}M_{11}\ddot{\varepsilon}_1^* + \frac{1}{8\ell^3}K_{11}\varepsilon_1^* + \frac{1}{8\sqrt{2}\ell^3}K_{12}\varphi = \frac{\sigma_1^*}{2} \\ \frac{1}{\sqrt{2}}M_{22}\ddot{\varphi} + K_{21}\varepsilon_1^* + \frac{1}{\sqrt{2}}K_{22}\varphi = 0, \end{cases} \tag{4.14}$$

where we define the uniaxial stress corresponding to $\varepsilon_1^*$ in a unit volume by $\sigma_1^* \equiv q/(4\sqrt{2}\ell^3)$. The definition is validated later through our investigation of the elastic modulus.

We consider a directional forced vibration by setting $\varepsilon_1^* \to \varepsilon_1^* \sin\omega t$. Under substitutions of the synchronized variables as $\varphi \to \varphi \sin\omega t$ and $\sigma_1^* \to \sigma_1^* \sin\omega t$, equation (4.14) simplifies, becoming

$$\left.\begin{array}{l} -\frac{1}{8\ell^3}M_{11}\omega^2\varepsilon_1^* + \frac{1}{8\ell^3}K_{11}\varepsilon_1^* + \frac{1}{8\sqrt{2}\ell^3}K_{12}\varphi = \frac{\sigma_1^*}{2} \\ -\frac{1}{\sqrt{2}}M_{22}\omega^2\varphi + K_{21}\varepsilon_1^* + \frac{1}{\sqrt{2}}K_{22}\varphi = 0. \end{array}\right\} \tag{4.15}$$

and

From the second equation of equation (4.15), the amplitude of $\varphi$ is arranged as a function of $\tilde{\omega}$ and $\varepsilon_1^*$,

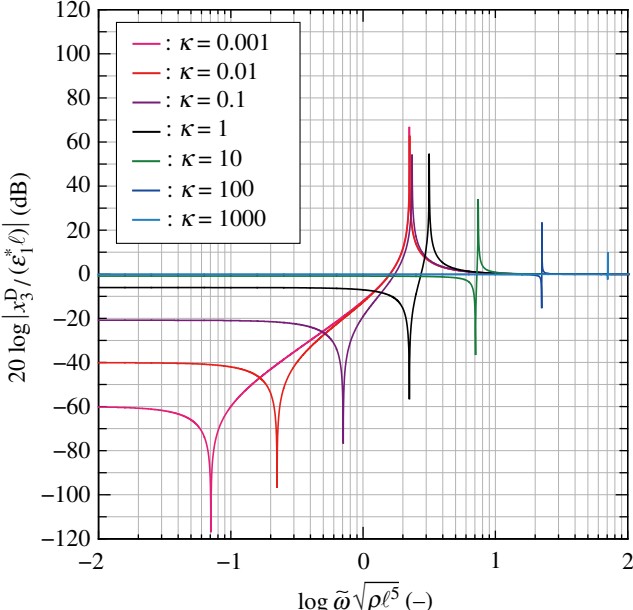

**Figure 8.** Frequency responses of the amplitude ratio of node A to node D, $\log \tilde{\omega}\sqrt{\rho \ell^5}$ versus $20 \log |x_3^D/(\varepsilon_1^* \ell)|$ using a base-10 logarithm scale, for $\kappa = 10^{-3}$–$10^3$.

that is,

$$\varphi = \frac{\sqrt{2}K_{21}}{M_{22}\omega^2 - K_{22}}\varepsilon_1^* = -\frac{1}{I_\varphi \tilde{\omega}^2 - (\kappa + 1)}\varepsilon_1^*. \tag{4.16}$$

Substituting equation (4.16) into equations (4.12), the amplitude ratios of $x_1^D$ and $x_3^D$ to the forced displacement are, respectively, given by

$$\frac{x_1^D}{\varepsilon_1^* \ell} = \frac{I_\varphi \tilde{\omega}^2 - (\kappa + 2)}{I_\varphi \tilde{\omega}^2 - (\kappa + 1)} \tag{4.17}$$

and

$$\frac{x_3^D}{\varepsilon_1^* \ell} = \frac{I_\varphi \tilde{\omega}^2 - \kappa}{I_\varphi \tilde{\omega}^2 - (\kappa + 1)}. \tag{4.18}$$

Figure 8 shows that the angular frequency responses of $H(\tilde{\omega}) \equiv x_3^D/(\varepsilon_1^* \ell)$ for the different values of $\kappa$, where the vertical axis indicates $20\log|H|$ [dB], using the base-10 logarithm scale for the number of decibels. Two peaks exist in figure 8 corresponding to the limit conditions for the two angular frequencies concerning the numerator and denominator terms of equation (4.18); we have

$$H \to 0 \quad \text{for } \tilde{\omega}_n = \sqrt{\frac{\kappa}{I_\varphi}} \tag{4.19}$$

and

$$H \to \infty \quad \text{for } \tilde{\omega}_d = \sqrt{\frac{\kappa + 1}{I_\varphi}}. \tag{4.20}$$

In equations (4.19) and (4.20), $\tilde{\omega}_n \to 0$ and $\tilde{\omega}_d \to I_\varphi^{-1/2}$ as $\kappa \to 0$, and $\tilde{\omega}_n = \tilde{\omega}_d$ as $\kappa \to \infty$. When $\kappa \to \infty$, $\varphi = 0$ regardless of the value of $\tilde{\omega}$ from equation (4.16); then $x_3^D = \theta \ell/\sqrt{2} = \varepsilon_1^* \ell$ and $H = 1$. In the high-frequency regime (figure 8), $H = 1$ for any $\kappa$, and represents the unimode transformation rotated through $\theta$. By contrast, from the red curves, the magnitude of $H$ decreases with decreasing $\kappa$ on the low-frequency side.

Substituting equation (4.16) into the first equation of equation (4.15), we find the stress–strain relation is expressed as

$$\sigma_1^* = \frac{k_D}{\ell}\left[\frac{1 - (2I_\theta\tilde{\omega}^2 - 1)(I_\varphi\tilde{\omega}^2 - (\kappa + 1))}{I_\varphi\tilde{\omega}^2 - (\kappa + 1)}\right]\varepsilon_1^* \equiv E^*(\tilde{\omega})\varepsilon_1^*. \tag{4.21}$$

R. Soc. Open Sci. **8**: 210768

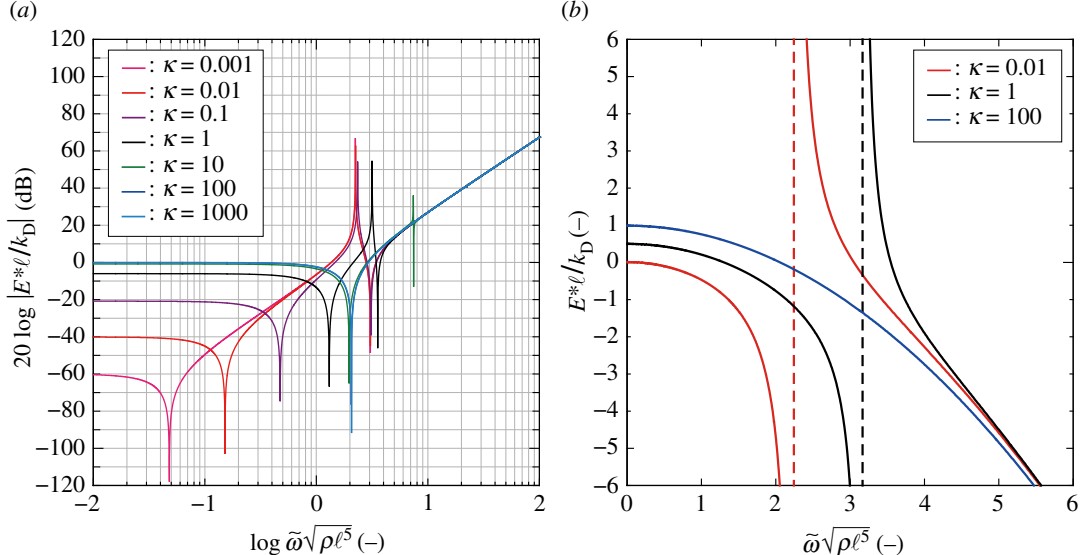

**Figure 9.** (a) Frequency response plots of $\log \tilde{\omega}\sqrt{\rho\ell^5}$ versus $20\log|E^*\ell/k_D|$ using a base-10 logarithm scale for $\kappa = 10^{-3}$–$10^3$; (b) an enlarged view of (a) for $\kappa = 10^{-2}$, 0 and $10^2$ using a linear scale.

Considering very low frequencies, $E^*(0) \to 0$ as $\kappa \to 0$ and $E^*(0) \to k_D/\ell$ as $\kappa \to \infty$. The former instance indicates a uniaxial vibration with zero-stiffness whereas the latter instance yields a force–displacement relationship $\sigma_1^*(4\ell^2) = 2k_D(2\varepsilon_1^*\ell) = 2k_D(2x_3^D)$ for the unimode transformation. The elastic relationship per unit cell validates the definition of $\sigma_1^*$.

Figure 9 shows the frequency responses of the non-dimensional elastic moduli $E^*(\tilde{\omega})\ell/k_d$ obtained by sweeping $\kappa$; specifically, (a) presents the log–log plots and (b) does an enlarged view of (a) using linear scales. The elastic modulus of the structural system for each $\kappa$ has three peaks described by equation (4.21); in detail, the two negative peaks are the solutions of the biquadratic equation in the numerator of equation (4.21), and the positive peak is consistent with equation (4.20). The very slow auxetic vibrations without elastic energy stored, mentioned in §4.1, are detected when $\kappa$ is sufficiently small, whereas high rigidity emerges under a fast vibration for any $\kappa$. The wide range for the stress–strain relation is reduced to low- and high-frequency modes, as shown in figure 7.

In figure 9b, when $\kappa = 10^2$, the stiffness begins with $E^*(0)\ell/k_D = 1$ and it decreases monotonically by increasing $\tilde{\omega}$—by passing through $E^* = 0$, the structure attains a resistance against higher angular frequencies with an anti-phase response. It seems that the depicted low-frequency behaviour with zero-stiffness is unstable in the linear vibration model. However, the bimode structure generates resistance like a solid under uniaxial finite transformation [30].

# 5. Conclusion

In summary, we extended the periodic tetrahedral structure developed in our previous study [30] to the kinematic model in the free vibration problem. We revisit the geometrical framework with corner-to-corner and edge-to-edge sharing of tetrahedra and the bimode transformation characterized by the two types of edge-sharing tetrahedral motions. We presented a modal analysis and frequency response analysis using the linear kinetic model and concluded that, with regard to the low-frequency and auxetic modes, the proposed structure possibly has a bifunctional vibration characteristic.

We constructed the unit-cell framework made up of eight regular tetrahedra with their point- and line-wise pivotal connections. We illustrated a one-eighth analytical model of the bimode structure incorporating interactions using three linear springs of differing spring constant. By introducing the nodal positions of the rotated tetrahedron, we identified the transformed unit cell producing auxeticity.

In preparation to solving the equations of motion for the bimode structure with relative spring constant $\kappa \equiv k/k_D$, where $k_B = k_C \equiv k$, we calculated the stiffness matrix $\mathbf{K}$ and the two rotational inertia $I_\theta$ and $I_\varphi$ with respect to $\theta$ and $\varphi$: the two parameters determine the distinct infinitesimal rotations of the tetrahedron unit.

From the equations of motion developed with $K$, $I_\theta$ and $I_\varphi$, the modal analyses yielded two distinct angular frequencies $\tilde{\omega}_{1,2}$. The limit operations revealed that $\tilde{\omega}_1 \to 0$ and $\tilde{\omega}_2$ becomes a finite constant as $\kappa \to 0$, whereas $\tilde{\omega}_1$ becomes a constant and $\tilde{\omega}_2 \to \infty$ as $\kappa \to \infty$. Correspondingly, an auxetic vibration mode with $\tilde{\omega}_1 = 0$ exists for $\phi_1$ and a structural vibration is locked geometrically for $\phi_2$. The self-contact problem can be resolved by an unclosed structure with $\theta_0 > 0$ applied in equation (2.11) that holds similar vibrational characteristics (high- and low-frequency modes) in the allowable range of $\theta_0$.

In regard to a uniaxial harmonic oscillation under a sinusoidal driving force, we calculated the amplitude ratio $H(\tilde{\omega}; \kappa)$ for representative nodal displacements and various elastic modulus $E^*(\tilde{\omega}; \kappa)$. This system supported a functional structure that has both low and high stiffness properties—low- and high-frequency modes obtained in the modal analysis are coupled—based on the unimode vibration with an isometric expansion that depends on $\kappa$.

Data accessibility. The data are available from the Dryad Digital Repository at https://doi.org/10.5061/dryad. 02v6wwq32 [33].

Authors' contributions. H.T. and S.A. performed the analyses and conducted the validation. H.T. and Y.S. designed the research. H.T. wrote the draft manuscript. All authors discussed the results and edited the manuscript. All authors gave final approval for publication and agree to be held accountable for the work performed therein.

Competing interests. We declare we have no competing interests.

Funding. The author (H.T.) gratefully acknowledges the financial support of the Japan Society for the Promotion of Science under a Grant-in-Aid for Scientific Research (B) (JSPS KAKENHI grant no. 18H01334).

Acknowledgements. We thank Richard Haase, PhD, from Edanz Group (https://en-author-services.edanz.com/ac) for editing a draft of this manuscript.

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
