## [Peer Review File · Royal Society Open Science]

Review History

RSOS-210768.R0 (Original submission)

Review form: Reviewer 1

Is the manuscript scientifically sound in its present form?

Yes

Are the interpretations and conclusions justified by the results?

Yes

Is the language acceptable?

Yes

Do you have any ethical concerns with this paper?

No

Have you any concerns about statistical analyses in this paper?

No

Recommendation?

Accept with minor revision (please list in comments)

Comments to the Author(s)

Observed typos:

page 5, line 35, formula (2.11) should have X_i

page 12, line 53, ... (a) presents ...

Review form: Reviewer 2

Is the manuscript scientifically sound in its present form?

Yes

Are the interpretations and conclusions justified by the results?

Yes

Is the language acceptable?

Yes

Do you have any ethical concerns with this paper?

No

Have you any concerns about statistical analyses in this paper?

No

Recommendation?

Accept as is

Comments to the Author(s)

The paper presents the auxetic vibration behaviour of periodic tetrahedral units with a shared edge. According to the reviewer's opinion, the paper is well-structured and clear. The topic is interesting and falls within the aim of the journal. In addition, the results are well-presented and could be helpful to further develop the same topic. Therefore, the paper can be accepted for publication in the current form.

Decision letter (RSOS-210768.R0)

Dear Dr Tanaka

On behalf of the Editors, we are pleased to inform you that your Manuscript RSOS-210768 "Auxetic vibration behaviours of periodic tetrahedral units with a shared edge" has been accepted for publication in Royal Society Open Science subject to minor revision in accordance with the

referees' reports. Please find the referees' comments along with any feedback from the Editors below my signature.

Please submit your revised manuscript and required files (see below) no later than 7 days from today's (ie 08-Sep-2021) date. Note: the ScholarOne system will 'lock' if submission of the revision is attempted 7 or more days after the deadline. If you do not think you will be able to meet this deadline please contact the editorial office immediately.

on behalf of Dr Adil Al-Mayah (Associate Editor) and R. Kerry Rowe (Subject Editor)
openscience@royalsociety.org

Reviewer comments to Author:
Reviewer: 1
Comments to the Author(s)
Observed typos:

page 5, line 35, formula (2.11) should have X_i

page 12, line 53, ... (a) presents ...

Reviewer: 2
Comments to the Author(s)
The paper presents the auxetic vibration behaviour of periodic tetrahedral units with a shared edge. According to the reviewer's opinion, the paper is well-structured and clear. The topic is interesting and falls within the aim of the journal. In addition, the results are well-presented and could be helpful to further develop the same topic. Therefore, the paper can be accepted for publication in the current form.

===PREPARING YOUR MANUSCRIPT===

===PREPARING YOUR REVISION IN SCHOLARONE===

- An individual file of each figure (EPS or print-quality PDF preferred [either format should be produced directly from original creation package], or original software format).
- An editable file of each table (.doc, .docx, .xls, .xlsx, or .csv).
- An editable file of all figure and table captions.

- Any electronic supplementary material (ESM).
- If you are requesting a discretionary waiver for the article processing charge, the waiver form must be included at this step.
- If you are providing image files for potential cover images, please upload these at this step, and inform the editorial office you have done so. You must hold the copyright to any image provided.
- A copy of your point-by-point response to referees and Editors. This will expedite the preparation of your proof.

- Ensure that your data access statement meets the requirements at <https://royalsociety.org/journals/authors/author-guidelines/#data>. You should ensure that you cite the dataset in your reference list. If you have deposited data etc in the Dryad repository, please only include the 'For publication' link at this stage. You should remove the 'For review' link.
- If you are requesting an article processing charge waiver, you must select the relevant waiver option (if requesting a discretionary waiver, the form should have been uploaded at Step 3 'File upload' above).
- If you have uploaded ESM files, please ensure you follow the guidance at <https://royalsociety.org/journals/authors/author-guidelines/#supplementary-material> to include a suitable title and informative caption. An example of appropriate titling and captioning may be found at https://figshare.com/articles/Table_S2_from_Is_there_a_trade-off_between_peak_performance_and_performance_breadth_across_temperatures_for_aerobic_scope_in_teleost_fishes_/3843624.

Author's Response to Decision Letter for (RSOS-210768.R0)

See Appendix A.

Decision letter (RSOS-210768.R1)

Dear Dr Tanaka,

I am pleased to inform you that your manuscript entitled "Auxetic vibration behaviours of periodic tetrahedral units with a shared edge" is now accepted for publication in Royal Society Open Science.

on behalf of Dr Adil Al-Mayah (Associate Editor) and R. Kerry Rowe (Subject Editor)
openscience@royalsociety.org

Appendix A

Professor A. Al-Mayah
Associate Editor

Professor R. Kerry Rowe
Subject Editor

September 15, 2021

RSOS-210768: Auxetic vibration behaviours of periodic tetrahedral units with a shared edge

Dear Professor Al-Mayah and Professor Rowe,

We appreciate the supportive reviews given by you and the reviewers.

We have revised the manuscript and corrected the typos pointed out by Reviewer #1. To make the paper more readable, we have made additional revisions, including those we had realized during the reviewing period. In the revised manuscript, the former changes to the text are marked **in blue**, and the latter are marked **in red** and are explained in detail below in Section (ii) covering general modifications.

We hope these changes are satisfactory to both you and the reviewers.

We also thank you in advance for your consideration of our revised manuscript.

Yours sincerely,

Hiro Tanaka, Shunki Asao, and Yoji Shibutani

Dr. Hiro Tanaka

Department of Mechanical Engineering
Osaka University
2-1 Yamadaoka, Suita, Osaka 565-0871, Japan
E-mail: htanaka@mech.eng.osaka-u.ac.jp; Tel: +81-6-6879-4120

(i) Point-by-Point Responses to Reviewer Comments

Below, we provide point-by-point responses to the reviewers' comments. In the revised manuscript, all changes are marked **in blue**.

Reviewer #1:

The paper investigates a periodic framework structure made of articulated regular tetrahedra. The framework is remarkable for several reasons: although over-constrained, it has two degrees of freedom for periodic deformations, the initial configuration has the eight tetrahedra of a unit cell positioned as a stellated octahedron and a particular auxetic deformation is singled out (unimode transformation). The structure is considered fitted (periodically) with three types of linear springs and is studied in relation to: stiffness and kinetic motion, vibrational characteristics and frequency response for uniaxial forced vibration. Analysis and results take very explicit form. This study extends related topics considered in [30].

*page 5, line 35, formula (2.11) should have X_i
page 12, line 53, ... (a) presents ...*

Response:

We are grateful for your supportive comments. The two typos pointed out by the reviewer were corrected in the revised manuscript.

Reviewer #2:

The paper presents the auxetic vibration behaviour of periodic tetrahedral units with a shared edge. According to the reviewer's opinion, the paper is well-structured and clear. The topic is interesting and falls within the aim of the journal. In addition, the results are well-presented and could be helpful to further develop the same topic. Therefore, the paper can be accepted for publication in the current form.

Response:

We are grateful for your supportive comments.

(ii) General Modifications Made by the Authors

After submission, we realized that some statements and mathematical descriptions should be improved to make this manuscript more readable. These modifications explained below did not change the structure, essential discussion or conclusions of the manuscript. In the revised manuscript, all changes are marked **in red**. We have also amended several symbolic errors and subtle expressions, which are not described below as they are of minor importance.

[Figures 7, 8, and 9]

In the original manuscript, we used the angular frequency variable $\tilde{\omega} = \omega/k_D \ell^2$ with a unit of $[\text{kg}^{-1/2} \cdot \text{m}^{-1}]$. Three figures (Figs. 7–9) have either a vertical or horizontal axis corresponding to $\tilde{\omega}$ and therefore are not uniquely determined by the diagrams. To regain uniqueness, we replaced $\tilde{\omega}$ with $\tilde{\omega} \sqrt{\rho \ell^5}$ in each of the figures (see below) and revised the manuscript accordingly.

Fig. 7 Relationship between $\tilde{\omega}$ and κ for the low and high angular frequency modes. The solid lines mark analytical solutions of Eq. (4.6), denoted $\tilde{\omega}_1$ and $\tilde{\omega}_2$ ($\tilde{\omega}_1 < \tilde{\omega}_2$). Note that $\tilde{\omega} \sqrt{\rho \ell^5} = \omega \sqrt{\rho \ell^3} / k_D$ is dimensionless.

Fig. 8 Frequency responses of the amplitude ratio of node A to node D, $\log \tilde{\omega} \sqrt{\rho \ell^5}$ vs $20 \log |x_3^D / (\varepsilon_1^* \ell)|$ using a base-10 logarithm scale, for $\kappa = 10^{-3} - 10^3$.

Fig. 9 (a) Frequency response plots of $\log \tilde{\omega} \sqrt{\rho \ell^5}$ vs $20 \log |E^* \ell / k_D|$ using a base-10 logarithm scale for $\kappa = 10^{-3} - 10^3$; (b) an enlarged view of (a) for $\kappa = 10^{-2}, 0$ and 10^2 using a linear scale.

[Equation (4.8)]

The eigenvector in Equation (4.8) represents a frequency mode of the structure and therefore should not have a unit in general. In the original manuscript, however, we expressed that

$$\phi_2 = \left(\frac{-I_\varphi}{\sqrt{2}I_\theta} \right) \#(4.8)$$

with units for the moment of inertia. In the revised manuscript, we have amended this non-dimensional description with the following;

$$\phi_2 = \left(-\sqrt{2} \frac{1}{(I_\theta/I_\varphi)} \right) \#(4.8)$$

In consequence of this change in Eq. (4.8), we have also recalculated the term in the last paragraph of Section 4(a) in page 11, and substituted the following result:

$$x_1^D = -\sqrt{2}\ell/12 \text{ and } x_3^D = 13\sqrt{2}\ell/12$$

[Equation (4.14)]

We made an error in defining the stress σ_1^* in Eq. (4.14). In the revised manuscript, we redefine the stress as $\sigma_1^* \equiv q/(4\sqrt{2}\ell^2)$. Then, the RHS of Eq. (4.14) and of (4.15) are revised as $\sigma_1^*/2$ instead of σ_1^* . The definition is validated later in the same section, which is why we have added the following statement after the definition.

“The definition is validated later through our investigation of the elastic modulus.”

The modification of the stress equation yields the correct elastic modulus in Eq. (4.21):

$$\sigma_1^* = \frac{k_D}{\ell} \left[\frac{1 - (2I_\theta\tilde{\omega}^2 - 1)(I_\varphi\tilde{\omega}^2 - (\kappa + 1))}{I_\varphi\tilde{\omega}^2 - (\kappa + 1)} \right] \varepsilon_1^* \equiv E^*(\tilde{\omega})\varepsilon_1^* \#(4.21)$$

Thus, all the relevant terms of $k_D/(2\ell)$ are replaced with k_D/ℓ in the main text and Figure 9. To justify this redefinition, we have added in page 13 in the revised manuscript, the following statement:

“..., whereas the latter instance yields a force-displacement relationship $\sigma_1^*(4\ell^2) = 2k_D(2\varepsilon_1^*\ell) = 2k_D(2x_3^D)$ for the unimode transformation. The elastic relationship per unit cell validates the definition of σ_1^* .”

[The 3rd paragraph of the introduction]

In the revised manuscript, to avoid misleading the readers, we have amended the statement referring to Ref. [27], which identified real auxetic materials that exhibit tetrahedral rotations.

“Recently, computations have predicted several types of polycrystal materials having negative values for the directional and/or the homogeneous Poisson ratios that arise through tetrahedral rotations [27].”

[Conclusions]

The structure we proposed in this study is locked physically in the higher vibration mode. However, our additional investigations after submission revealed that this vibration-contact problem can be avoided by considering the configuration of the unclosed structure, initially rotated by $\theta_0 > 0$. During modelling, the structure holds similar vibrational characteristics with those of the original closed structure (see Fig. S1 below).

Fig. S1: Vibration characteristics ($\tilde{\omega}$ vs κ) of the unclosed structures with $\theta_0 > 0$, obtained in multi-body dynamics simulations. Within $0^\circ < \theta_0 < \pi/2 - 1/\sqrt{2} \approx 49.6^\circ$, high or low frequency modes have no interferences through self-contact of components. The inset shows an example with $\theta_0 = 20^\circ$.

In analysing the mechanism, the plots were generated during numerical simulations using multi-body dynamics software (Adams[®], MSC Software Corp.). Note that the numerical points for $\theta_0 = 0$, indicated by open circles, agree with theoretical predictions (dashed curves). Following these latest findings, we have revised the manuscript adding a statement in the 4th paragraph of Conclusions;

“The self-contact problem can be resolved by an unclosed structure with $\theta_0 > 0$ applied in Eq. (2.12) that holds similar vibrational characteristics (high and low frequency modes) in the allowable range of θ_0 .”

[Data Accessibility]

We have modified the data accessibility statement and citation as follows:

“The data are available from the Dryad Digital Repository at <https://doi.org/10.5061/dryad.02v6wwq32> [33].”

33. Tanaka H, Asao S, Shibutani Y. 2021 Data from: Auxetic vibration behaviours of periodic tetrahedral units with a shared edge. Dryad Digital Repository. ([10.5061/dryad.02v6wwq32](https://doi.org/10.5061/dryad.02v6wwq32))